# 2D Hierarchical NiMoO_4_ Nanosheets/Activated Carbon Nanocomposites for High Performance Supercapacitors: The Effect of Nickel to Molybdenum Ratios

**DOI:** 10.3390/ma16031264

**Published:** 2023-02-01

**Authors:** Esraa Hamdi, Abdalla Abdelwahab, Ahmed A. Farghali, Waleed M. A. El Rouby, Francisco Carrasco-Marín

**Affiliations:** 1Materials Science and Nanotechnology Department, Faculty of Postgraduate Studies for Advanced Sciences, Beni-Suef University, Beni-Suef 62511, Beni-Suef, Egypt; 2Faculty of Science, Galala University, Sokhna 43511, Suez, Egypt; 3Carbon Materials Research Group, Faculty of Science, University of Granada, 18071 Granada, Spain

**Keywords:** activated carbon, nickel molybdate, hierarchical structure, nanocomposite, supercapacitors

## Abstract

Supercapacitors have the potential to be used in a variety of fields, including electric vehicles, and a lot of research is focused on unique electrode materials to enhance capacitance and stability. Herein, we prepared nickel molybdate/activated carbon (AC) nanocomposites using a facile impregnation method that preserved the carbon surface area. In order to study how the nickel-to-molybdenum ratio affects the efficiency of the electrode, different ratios between Ni-Mo were prepared and tested as supercapacitor electrodes, namely in the following ratios: 1:1, 1:2, 1:3, 1:4, and 1:5. X-ray diffraction, X-ray photoelectron spectroscopy, FESEM, HRTEM, and BET devices were extensively used to analyze the structure of the nanocomposites. The structure of the prepared nickel molybdates was discovered to be 2D hierarchical nanosheets, which functionalized the carbon surface. Among all of the electrodes, the best molar ratio between Ni-Mo was found to be 1:3 NiMo3/AC reaching (541 F·g^−1^) of specific capacitance at a current density of 1 A·g^−1^, and 67 W·h·Kg^−1^ of energy density at a power density of 487 W·Kg^−1^. Furthermore, after 4000 repetitive cycles at a large current density of 4 A·g^−1^, an amazing capacitance stability of 97.7% was maintained. This remarkable electrochemical activity for NiMo3/AC could be credited towards its 2D hierarchical structure, which has a huge surface area of 1703 m^2^·g^−1^, high pore volume of 0.925 cm^3^·g^−1^, and large particle size distribution.

## 1. Introduction

The excessive usage of coal and oil as an energy source has caused many environmental disasters such as global warming. Therefore, renewable energies are considered to be the best alternatives for fossil fuels [1,2]. Renewable energy systems such as solar cells, fuel cells, and wind turbines are important systems for energy generation [3]. Consequently, if the energy produced from a renewable energy system is not used at the main time of its generation, it will dissipate. So, it is mandatory to store this produced energy in energy storage systems such as batteries or supercapacitors [4]. Batteries suffer from several drawbacks such as high self-discharge rates, a large size, high toxicity, low power density, and flammability [4,5]. Compared with batteries, supercapacitors have many features that allow it to outperform other energy applications, such as a long durability power density, higher rates of charge and discharge, and being environmental friendly. Supercapacitors are divided into two kinds based on how they store energy: electrochemical double-layer capacitors (EDLCs) and pseudocapacitors [6,7]. The mechanism of the first type of energy storage is based on the adsorption of ions into the carbon material surface. The mechanism of the second type of energy storage depends on the reversible redox reactions between the electrode and electrolyte, similar to that in transition metal oxides or conducting polymers.

In order to gain the advantages of both EDLCs and pseudocapacitors, hybrid supercapacitors emerged [8,9]. A hybrid supercapacitor consists of either a carbon material and transition metal oxide or a polymeric material [10,11,12,13]. It has a non-flammable nature, a very stable electrical system, and a large energy and power density. Hence, successfully designing electrode materials between carbon and ultrathin two-dimensional (2D) mesoporous transition metal oxides would have a great impact on high-performance supercapacitor technology. Among the different carbon materials, biomass-derived activated carbon is a well-known carbon material in supercapacitor applications because of its large surface area, as well as good thermal and chemical stability [14]. Moreover, metal molybdates have piqued the interest of researchers considering supercapacitors because of their exceptional electro-chemical performance and low cost [15].

According to research, nanostructures and electrodes with diameters smaller than 10 nm would be advantageous and useful for the next generation of supercapacitors, as pseudocapacitors store the charges in close to the surface of the nanomaterials [16]. Consequently, the recent supercapacitor technology has promoted ultrathin mesoporous nanostructured materials. The benefits of the preparation of such electrodes are as follows: an increase in electrochemical performance with a decrease in particle sizes, as it provides more active sites due to the high surface areas, which results in more ion adsorption and redox reactions. Transition metal oxides have a great impact on the performance of supercapacitors. Nickel molybdate, NiMoO_4_, is one of the metal oxides that has a promising electrochemical performance. Many researchers have given great attention to studying the electrochemical properties of NiMoO_4_ and have added many modifications to improve its performance.

For example, A. Ajay et al. [17] prepared 2D amorphous NiMoO_4_ nanoflakes and tested them as supercapacitors electrodes. The 2D amorphous NiMoO_4_ nanoflakes electrodes exhibited a specific mass capacitance of 1650 F·g^−1^ with an energy density of 92 WhKg^−1^ at a power density of 23 KWKg^−1^. In another work, by D. Cai et al. [16], hierarchical mesoporous NiMoO_4_ nanosheets were synthesized for supercapacitors; here, 1200.5 F·g^−1^ was the value of the specific capacitance that produced 20 A·g^−1^ of current density with a capacitance retention of around 75%. This lower capacitance retention in the case of pseudocapacitors can be developed through their formulation with the carbon matrix. O. Rabbani et al. [9] prepared a manganese-nickel molybdate/reduced graphene oxide composite using the sonochemical assisted method, where 161.1 mA·h·g^−1^ was the specific capacitance that was obtained at 2 A·g^−1^ current density. In another study, a NiMoO_4_ nanofiber with a graphitic carbon nitride nanocomposite was prepared in order to obtain a specific capacitance of 510 F·g^−1^ at 1 A·g^−1^, and it showed stability up to 91.8% after 2000 cycles and an energy density of 11.3 Whkg^−1^ [18]. To the best of our knowledge, the effect of the ratio between Ni-Mo on the electrochemical performance of NiMoO_4_ had not been studied before.

So, the objectives of this work were as follows: to (i) study the effect of molar ratios between Ni and Mo on the electrochemical performance of NiMoO_4_ and to (ii) design a NiMoO_4_/activated carbon electrode as a symmetric supercapacitor with an enhanced stability and high energy density. In this work, hybrid supercapacitor electrodes consisting of nickel molybdate and activated carbon (NiMoO_4_/AC) were fabricated using different molar ratios between Ni and Mo, namely; 1:1, 1:2, 1:3, 1:4, and 1:5. Biomass-derived activated carbon was prepared from the argan seed shells in order to obtain a high surface area carbon material. Argan seed was chosen as it produces carbon with a large surface area and enhanced volume of micro and meso pores, in addition to the excellent electrochemical properties of activated carbon that are obtained from argan seed. A chemical activation process with KOH was used to enhance the obtained carbon’s surface area. The impregnation method was used to functionalize the activated carbon with NiMoO_4_ for preserving the activated carbon’s surface area. The prepared materials were tested as symmetric supercapacitors electrodes in the acidic media of 1 M H_2_SO_4_.

## 2. Experimental Section

### 2.1. Preparation of Activated Carbon (AC)

The activated carbon was prepared using argan seed shells. The argan shells were carbonized in a N_2_ environment at 300 °C for 2 h at a heating rate of 5 °C/min. The prepared carbon material was activated with KOH (85%) at a mass ratio for carbon and KOH of 1:2. After that, the mixture was heated again in N_2_ flow for 2 h at 800 °C. Finally, 0.1 N HCl was used to wash the produced carbon, and then distilled water was used until no chloride ions were detected in the filtrate of the 0.1 N AgNO_3_ solution.

### 2.2. Preparation of NiMoO_4_/AC

The NiMoO_4_/AC nanocomposites were prepared using the impregnation method [19]. Different molar ratios between nickel and molybdenum (Ni-Mo) were prepared, namely, 1:1, 1:2, 1:3, 1:4, and 1:5, and were denoted as NiMo1/AC, NiMo2/AC, NiMo3/AC, NiMo4/AC, and NiMo5/AC, respectively. The total weight percentage of nickel molybdate (NiMoO_4_) to activated carbon was calculated to be 1% and was fixed in all of the samples. For example, for the preparation of NiMo1/AC, 3.39 mg of Ni(CH_3_COO)_2.4_H_2_O and 3.2 mg of pure molybdenum (Mo) were mixed in 3 mL of distilled water, and they were then sonicated for 15 min. The obtained mixture was homogenously added to 300 mg of activated carbon, drop by drop, and then dried by microwave. Afterwards, the obtained powder was treated at 700 °C for 2 h in N_2_. The other compounds, i.e., NiMo2/AC, NiMo3/AC, NiMo4/AC, and NiMo5/AC, were prepared using the same procedure, except with changing the wt. ratios between Ni and Mo. The prepared materials together with their Ni-Mo ratios are compiled in Table 1.

### 2.3. Characterization of Materials

The prepared samples were analyzed using different physicochemical characterization tools. X-ray diffraction (XRD) (PANalytical Empyrean diractometer, with CuK α radiation, wavelength λ = 1.54060 Å, accelerating voltage 40 KV, current 30 mA, start and end position of 5–80°, and scan step 0:05°) was used to determine the crystal structure of the samples. Raman spectroscopy was investigated by Bruker Senterra Raman Microscope (Bruker Optics Inc., Ettlingen, Germany). TriStar II 3020, Micromeritics, USA, was utilized to study the pore size distributions and the surface area of N_2_ adsorption and desorption at −196 °C. The morphology and crystal structure were investigated by field emission scanning electron microscopy (SEM, Zeiss sigma VP 500, Oberkochen, Germany) and high-resolution transmission electron microscopy (HRTEM, JEOL JEM 2100, Tokyo, Japan). Energy dispersive X-ray (FESEM-EDX) analysis and elemental mapping were carried out using SEM (Zeiss sigma VP 500, Oberkochen, Germany). The oxidation states were characterized by X-ray photoelectron spectroscopy (K-ALPHA, Thermo Fisher Scientific, Waltham, MA, USA) with monochromatic X-ray Al K-alpha radiation of 10 to 1350 eV, spot size of 400 µm, and a pressure range of 10^−9^ mbar with full-spectrum pass energy of 200 eV, and a narrow-spectrum of 50 eV.

### 2.4. Electrochemical Measurement

The potentiostat/galvanostat (Autolab/PGSTAT 302N) in two-electrode configuration was used to investigate the electrochemical performance. The working electrodes became ready after mixing 90 wt% and 10 wt% of a sample and polyvinylidene difluoride (PVDF), respectively, with a mass loading of 5 mg for each electrode. Then, the slurry was pressed onto graphitic substrates with diameters of 9 mm, and dried at 80 °C for 12 h. The activity of the electrodes was tested in 1 M H_2_SO_4_ using galvanostatic charge-discharge (GCD), cyclic voltammetry (CV), and electrochemical impedance spectroscopy (EIS). The cyclic voltammetry (CV) was performed within a potential window range of 0–1 V at a different scan rate range of 5–10 mV·s^−1^. GCD measurements were measured in the same potential window at a current density range of 0.25–10 A·g^−1^. Furthermore, EIS was investigated within the frequency range of 10 mHz–100 KHz.

## 3. Results and Discussion

### 3.1. Structural and Morphological Analysis

Figure 1 presents the XRD patterns of NiMo1/AC, NiMo2/AC, NiMo3/AC, NiMo4/AC, and NiMo5/AC. An amorphous structure of nickel molybdate/activated carbon nanocomposite was obtained with a broad peak at 2θ of 44°, which was related to the (101) plane of carbon [20]. The diffraction peaks at 2θ of 40.7° and 47° were assigned to the (110) and (111) planes of NiMoO_4_ (JCPDS card 13-0128) [21]. It is also noted that there was an overlap between the carbon peak at 2θ of 44° and the NiMoO_4_ peak at 2θ of 47°. The absence of any contaminated peaks confirmed the successful preparations of pure composites between NiMoO_4_ and activated carbon.

Figure 2 shows the pore size distributions for the as prepared samples obtained from the corresponding N_2_ adsorption/desorption isotherms. Figure 2a shows the pore size distribution for AC, in which the majority of the pores are in the micropore range of <2 nm. In Figure 2, it can be seen that all of the samples have a pore size distribution range of 0.4 to about 2.0 nm, except for NiMo3/AC, in which the pore size distribution ranges from 0.6 to 2.7 nm. These data show that all of the samples had pores in the micro-pore range, except for NiMo3/AC, which had a broad range of pore sizes in the micro and mesoporous sizes. Micro/mesopores are important in supercapacitor applications, where the micropores offer a high surface area, which in turn increases the number of ionic adsorptions, while the mesopores facilitate the transport of ionic species to reach the micropores [22]. Table 2 summarises the results of the N_2_ adsorption isotherms for the as-prepared samples.

From Table 2, it can be seen that all of the samples had a comparable specific surface area (S_BET_) and mean pore diameter ranging from 1043 m^2^·g^−1^ to 1703 m^2^·g^−1^ and 1.07 to 1.76 nm, respectively. The largest surface area and pore volume were found in NiMo3/AC. These findings were also supported by the nonlocal density functional theory (NLDFT), which is characteristic for amorphous microporous materials to the N_2_-isotherms (Table 2). All of the samples had high pore volumes that were enhanced in NiMo3/AC, which reached 0.925 cm^3^·g^−1^. As can be seen from Table 2, the surface area of the activated carbon was preserved by using the impregnation method for functionalizing the activated carbon with nickel molybdate. Eldeep et al. [23] prepared carbon xerogel with 701.9 m^2^·g^−1^ for the high surface area, but after doping with NiCo_2_O_4_ through the hydrothermal process this high surface area decreased to 64.0 m^2^·g^−1^. The optimization of the pore size could lead to the maximum amount of energy being stored in the nanoporous materials during supercapacitor applications.

Figure 3 and Figure 4 show the XP spectra for NiMo1/AC and NiMo3/AC, respectively, as a comparison between both samples. Figure 3a shows a survey spectrum of NiMo1/AC, which includes bands for C 1s, Ni 2p, Mo 3d, and O 1s at 285.4 eV, 856.39 eV, 233.5 eV, and 532.4 eV, respectively [18]. In addition, Figure 3b shows the peak deconvolutions for C 1s at 284.5 eV, 285.8 eV, and 287.2 eV, which refer to bands C-C/C=C, C-O, and C=O, respectively. In Figure 3c, the Ni peaks at 855.6 and 873.5 eV are ascribed as Ni 2p_3/2_ and Ni 2p_1/2_, respectively. The satellite peaks for Ni^+2^ and Ni^+3^ appeared at 862.1 eV and 866.63 eV, respectively [20,24]. Figure 3d shows the main peaks of Mo^+6^ levels, where 232.4 eV refers to Mo 3d_5/2_ and 235.5 eV refers to Mo 3d_3/2_ with an energy band gap of 3.1 eV [6,25]. The peak deconvolutions for O 1s are represented in Figure 3e, where the peaks at 530.7 eV, 531.6 eV, and approximately 533 eV refer to bands between the metal and oxygen (M-O-M), C=O, and C-O, respectively.

Figure 4a shows the survey spectrum of NiMo3/AC, which include bands for C 1s, Ni 2p, Mo 3d, and O 1s at 285.4 eV, 857.15 eV, 233.5 eV, and 532.4 eV, respectively. Figure 4b shows the peaks for C 1s at 284.5 eV, 285.8 eV, and 287.2 eV, which refer to the bands C-C/C=C, C-O, and C=O. Figure 4c shows the Ni 2p peaks, which are located at 856.3 eV and 874.2 eV and refer to the main peaks at Ni 2p_3/2_ and Ni 2p_1/2_, respectively. The value for the spin-orbit splitting for the main peak core level is 17.9 eV. The satellite peaks that appear at 862.1 eV and 864.3 eV refer to Ni^+2^ and Ni^+3^, respectively. Figure 4d shows the main peaks of the Mo^+6^ levels at 232.5 eV and 235.6 eV, which refer to Mo 3d_5/2_ and Mo 3d_3/2_, respectively, with a band gap energy of 3.1 eV. Figure 4e shows the O 1s peaks at 530.7 eV, 531.6 eV, and 232.7 eV, which refer to the bands between the metal and oxygen (M-O-M) and carbon-oxygen functionalities (C=O and C-O), respectively. The results from Figure 3 and Figure 4 confirm the successful functionalization of activated carbon with 2D nickel molybdate nanosheets. Moreover, the full width at half maximum (FWHM) for the C-C peaks of NiMo1/AC and NiMo3/AC were calculated to be 1.30 eV and 1.33 eV, respectively. This proves that NiMo3/AC had a higher conductivity because an increase in the FWHM value corresponds to a decrease in resistance due to the development of graphitic clusters around the metals [22].

Figure 5 shows the SEM images for NiMo1/AC, NiMo2/AC, NiMo3/AC, NiMo4/AC, and NiMo5/AC. A uniform morphology of the 2D hierarchical structure of the NiMoO_4_ nanosheets was observed for all of the samples decorated on the irregular shape of the activated carbon. The morphology of the 2D hierarchical nanosheets for NiMoO_4_ is formed because its lateral size is much larger than its thickness [16]. In order to ensure the exact elements of the prepared composites, EDX and elemental mapping were performed for NiMo3/AC and the image are shown in Figure 6. Figure 6a shows the selected area for the characterization of NiMo3/AC, Figure 6b shows the ratio between the elements where the ratio between Ni-Mo is confirmed, and Figure 6c shows the elemental mapping distribution. As can be seen from Figure 6, there was a homogenous distribution of elements throughout the whole matrix. In addition, the exact stoichiometry between Ni and Mo was achieved in an excellent way.

The morphology and microstructure of the samples were characterized by HRTEM, as presented in Figure 7. The HRTEM images of NiMo1/AC, NiMo2/AC, NiMo3/AC, NiMo4/AC, and NiMo5/AC are shown in Figure 7a–e, respectively. The images in Figure 7a,b clearly reveal the ultrathin hierarchical nanosheets over the activated carbon surface [16,26]. Moreover, a second morphology of nanoparticles was observed and homogenously distributed onto the carbon surface. The particles sizes ranged from 1.8 nm for NiMo2/AC to 4.2 nm for NiMo4/AC. With a lower magnification, Figure 7c–e shows the graphitic lines, which confirms the good conductivity of the prepared electrodes. The lattice spacing of carbon is about 0.32 nm, which can be attributed to the interlayer distance of the graphene sheets.

Raman analysis was performed for the prepared samples in order to evaluate the order and graphitic natures of the samples, as shown in Figure 8. Two strong peaks for the D-band and G-band of carbon are observed at about 1346 cm^−1^ and 1599 cm^−1^, respectively. The D-band occurred due to the alternating benzene ring vibrations, while the G-band developed due to the sp^2^ structure [27]. A reduction in the D- and G-bands was shown for the NiMo2/AC sample, suggesting a more disordered structure, which is in agreement with the surface area data.

### 3.2. Electrochemical Study

Figure 9a shows the curves of the CV for all of the samples in the 1 M H_2_SO_4_ electrolyte at a scan rate of 20 mV·s^−1^ in a potential window range of 0 to 1 V. In Figure 9a, a development in the cyclic voltammograms area was noticed up until functionalizing the AC with nickel molybdate. All CV curves formed a near-rectangular shape, which indicates that all of the samples had an ideal electrical double-layer capacitive behavior. The highest CV area was found for NiMo3/AC, which suggests a higher capacitance for this composite. It is also obvious that the redox peaks for NiMoO_4_ were absent due to the two electrode systems. Figure 9b displays curves of the CV for NiMo3/AC at various scan rates. The total cell capacitance was calculated according to Equation (1) and the specific capacitance of the samples was calculated according to Equation (2) [22].
(1)$$C=\frac{\int \mathrm{Idv}}{ʋ\Delta V}$$
(2)$$c_{s}=4\times \frac{I\times \Delta t}{m\times \Delta V}$$
where the integrated area under the CV curve is (ʃIdv), ʋ refers to the scan rate, the potential window is ∆V, and m is the mass of the loading active material for the two electrodes. Factor 4 is used to compute the total cell capacitance of two electrodes to reward the capacitance of the single electrode. Table 3 depicts the calculated specific capacitance for all of the electrodes from the cyclic voltammetry measurements at various scan rates and current densities.

Among all of the electrodes, the best performance was provided by NiMo3/AC, which had the highest specific capacitance. This could be because of its high surface area and pore volume, which allowed for faster charge transport and diffusion. The BET analysis revealed that the pore diameter ranged from 1.07 nm to 1.76 nm. Such a wide pore diameter is advantageous, considering the fact that the ionic radius for sulfate ions is 0.242 nm, which can easily be adsorbed into these pores, thereby improving the overall capacitance [17].

The values of the specific capacitances range from 172 F·g^−1^ for AC to 454 F·g^−1^ for NiMo3/AC at a scan rate of 5 mV·s^−1^. This improvement in the specific capacitance could be attributed to the cumulative effect of both the electrical double-layer and faradic capacitances at the bulk and the surface of these electrodes. Moreover, when increasing the scan rates, the specific capacitance values are decreased. The reason for such behavior is the limited entry of ions into the deeper pores, which results in lower available active site participation for charge storage at a high scan rate.

The galvanostatic charge-discharge of all electrodes for NiMo1/AC, NiMo2/AC, NiMo3/AC, NiMo4/AC, and NiMo5/AC at 1 A·g^−1^ of current density are shown in Figure 10a. The best performing electrode was NiMo3/AC, which achieved a specific capacitance of 541 F/g^−1^ at 1 A/g^−1^ current density. The calculated specific capacitances for NiMo1/AC, NiMo2/AC, NiMo3/AC, NiMo4/AC, and NiMo5/AC at current densities of 0.5, 0.75, and 1 A·g^−1^ are depicted in Table 3. Figure 10b displays the GCD of NiMo3/AC at different current densities of 0.5, 0.75, 1, 5, and 10 A·g^−1^. A very small voltage (*IR*) drop was noticed at different current densities, indicating a lower pore resistance, which facilitated the redox reactions on the electrode surface. Figure 10c shows the change in specific capacitance with various current densities of 0.25, 0.5, 0.75, 1, 5, and 10 A·g^−1^ for NiMo1/AC, NiMo2/AC, NiMo3/AC, NiMo4/AC, and NiMo5/AC.

Figure 11a represents the Nyquist plots for AC, NiMo1/AC, NiMo2/AC, NiMo3/AC, NiMo4/AC, and NiMo5/AC measured within the frequency range of 10 mHz–100 kHz. The capacitive nature and ohmic behavior are shown by the imaginary (Z″) and real components (Z′), respectively. The inset of Figure 11a shows the equivalent circuit fit diagram, which was obtained using Nova 1.11 software. The fitting data for the electrolytic resistance (Rs) and the electrode equivalent series resistance (ESR) are represented in Table 3. From Figure 11a, as the frequency decreased from a higher to lower range, the deeper pores became available for ionic adsorption, which is known as the Warburg region (W) [16,17]. In the high frequency range, semicircles appeared in the real part, which represents the electrons transfer between the electrode and the electrolyte, which is known as the charge transfer resistance (R_ct_). The semicircle diameter represents the resistance of the electrode material, the smaller the diameter of the semicircle, the lower its resistance. This in turn gives a higher conductivity and higher specific capacitance. From Table 3, NiMo3/AC had the smallest equivalent series resistance (ESR (1.6 Ω)), which is an indication of a better conductivity and higher charge transfer. The straight line in the electrochemical impedance spectroscopy (EIS) indicated the electric double layer capacitive performance of the electrodes, this straight line appeared in a small range of frequencies in the imaginary part. The slope of the straight line indicated the electrode’s specific capacitance. The lower the slope, the higher specific capacitance obtained due to the lower electrode resistance. Figure 11b shows the stability of electrodes after 4000 cycles at 4 A·g^−1^ of current density. All electrodes exhibited an excellent capacitance retention (RC) after 4000 cycles, ranging between 91.79% for NiMo4/AC to 98.8% for NiMo2/AC, as shown in Table 3.

The energy density is an important parameter in supercapacitor applications, so the following equations were used to calculate the energy density and power density for the papared materials [22]:(3)$$E\left({Wh\mathrm{Kg}}^{-1}\right)=\frac{C_{s}\left({\mathrm{Fg}}^{-1}\right)\times \Delta V^{2}\left(V\right)}{2\times 3.6}$$
(4)$$P\left({W\mathrm{Kg}}^{-1}\right)=\frac{1}{2}\times \frac{I\left(A\right)\times \Delta V\left(V\right)}{m\left(\mathrm{Kg}\right)}$$

Figure 11c depicts a ragone plot, which represents the relationship between the energy and power densities for all of the prepared materials. NiMo3/AC released 67 W h Kg^−1^ as the maximum value for energy density at 487 W Kg^−1^ of power density. At the beginning, the energy density increased with increasing the molar ratios between Ni-Mo from 1:2 to 1:3; after that, it started to decrease again due to the reduction in the specific capacitance. This trend could be explained by the lower energy loss caused by a decrease in the equivalent series resistance (ESR) together with better ionic diffusion through the NiMo3/AC network. This trend in capacitance change with Ni-Mo ratios is expressed in Figure 11d.

In addition, when comparing these data with the already published data for NiMoO_4_ nanostructures, in Table 4, a higher capacitance and electrode retention were obtained for NiMo3/AC. The proposed future work will study the effect of different concentration of Ni in the preparation of nickel molybdate nanoparticles considering their activity. These materials can be decorated on the surface of carbon quantum dots or the nanodiamond matrix in order to enhance the performance of these materials in supercapacitor applications.

## 4. Conclusions

A simple impregnation method was developed for functionalizing activated carbon with nickel molybdate in order to preserve the surface area. Different molar ratios were used between Ni-Mo in the nanocomposite preparation. The prepared NiMoO_4_ had the morphology of 2D nanosheets, which enhanced the charge transport inside the composites. The optimum ratio between Ni and Mo was found to be 1:3; NiMo3/AC, which delivered high specific capacitances of about 541 F·g^−1^ at 1 A·g^−1^ and excellent electrodes stabilities that reached 97.7% after 4000 repetitive cycles. Moreover, the NiMo3/AC sample provided a significant energy density of about 67 W h Kg^−1^ at 487 W Kg^−1^ of power density. These high specific capacitances and electrodes stabilities could be attributed to its high specific surface areas, large pore volumes, and large pore size distributions. In addition, the EIS measurements showed that the lowest equivalent series resistance was for NiMo3/AC, indicating lower resistances and higher charge transfers. These results suggest that there is an optimum molar ratio between Ni and Mo in the synthesis of nickel molybdate nanoparticles, which greatly affects their performance in energy applications such as supercapacitors.

## Figures and Tables

**Figure 1 materials-16-01264-f001:**
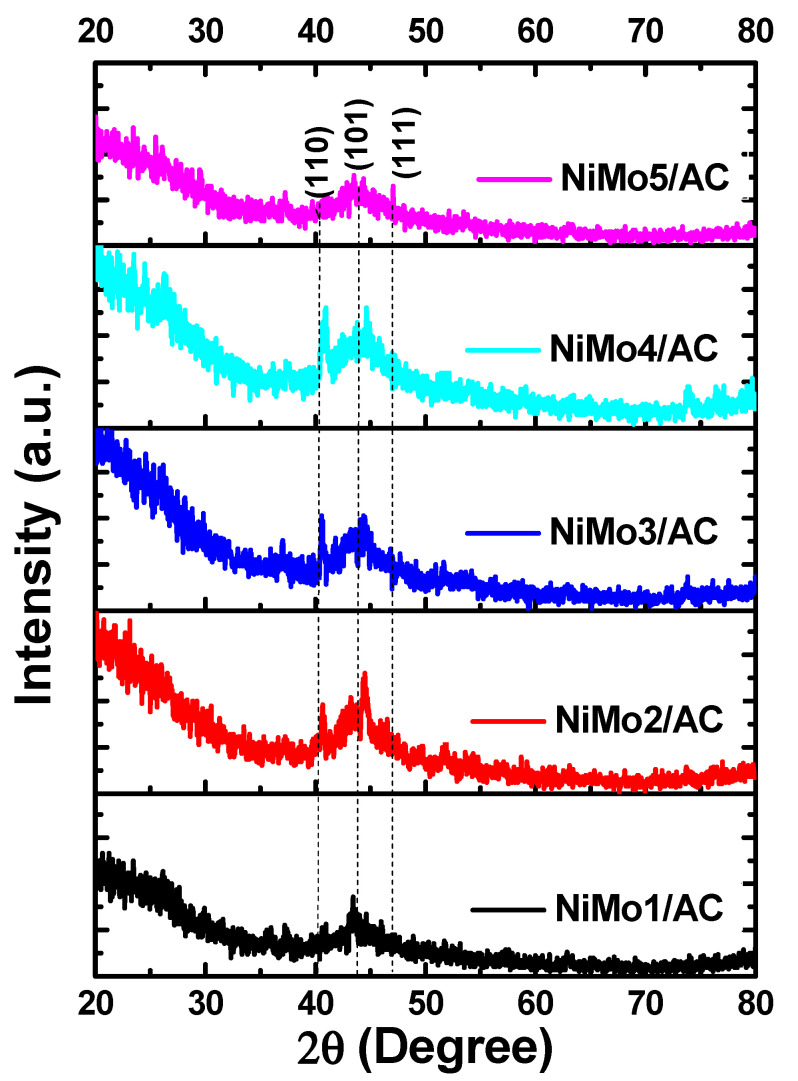
XRD patterns for the prepared composites.

**Figure 2 materials-16-01264-f002:**
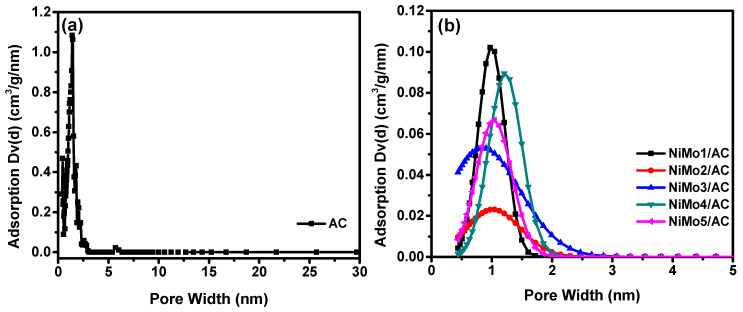
Pore size distribution for (**a**) AC and (**b**) NiMo1/AC, NiMo2/AC, NiMo3/AC, NiMo4/AC, and NiMo5/AC from N_2_ adsorption-desorption isotherms at −196 °C.

**Figure 3 materials-16-01264-f003:**
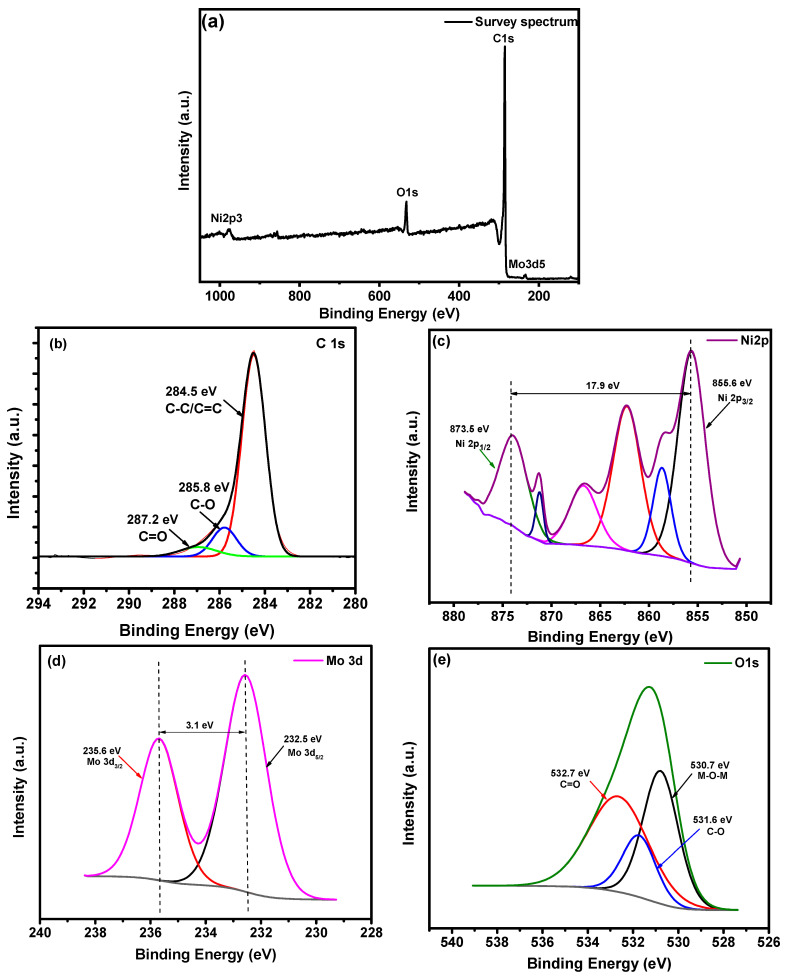
XPS spectra of NiMo1/AC: (**a**) Survey spectrum; (**b**–**e**) core-level spectra (**b**) C 1s, (**c**) Mo 3d, (**d**) Ni 2p, and (**e**) O 1s.

**Figure 4 materials-16-01264-f004:**
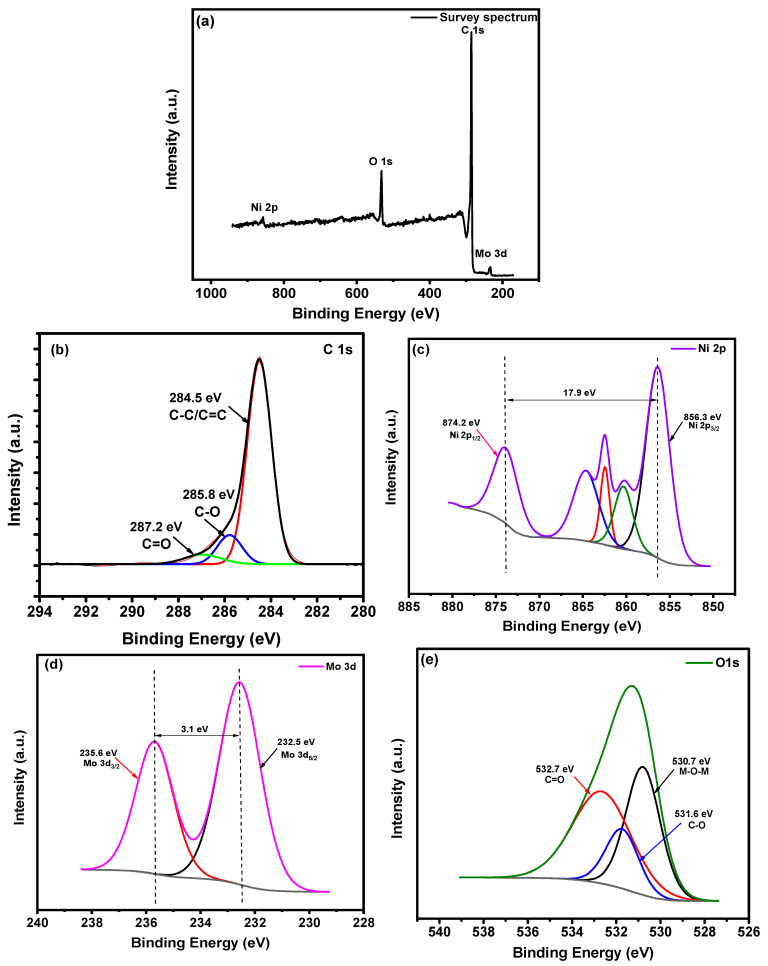
XPS spectra of NiMo3/AC: (**a**) survey spectrum; (**b**–**e**) core-level spectra (**b**) C 1s, (**c**) Mo 3d, (**d**) Ni 2p, and (**e**) O 1s.

**Figure 5 materials-16-01264-f005:**
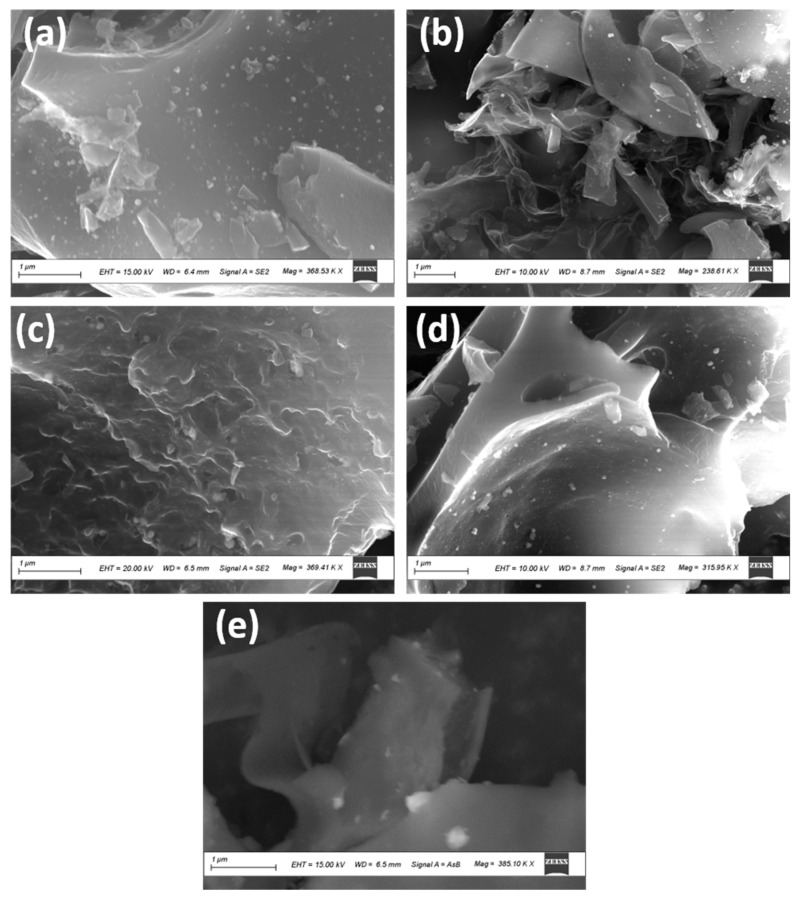
SEM images of samples (**a**) NiMo1/AC, (**b**) NiMo2/AC, (**c**) NiMo3/AC, (**d**) NiMo4/AC, and (**e**) NiMo5/AC.

**Figure 6 materials-16-01264-f006:**
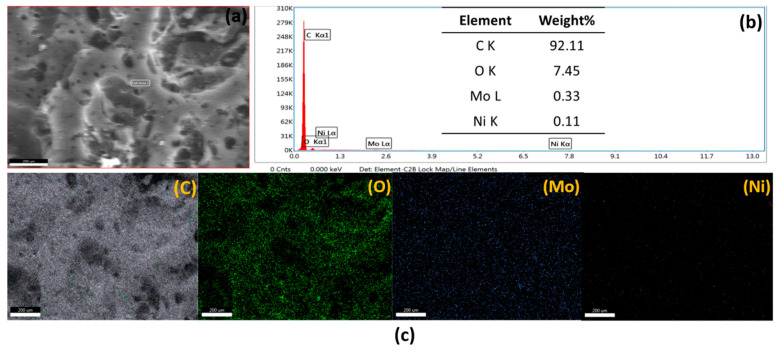
FESEM-EDX elemental mapping analysis for NiMo3/AC. (**a**) The full area of elemental mapping (**b**) EDX analysis (**c**) Elemental distribution of C, O, Mo and Ni.

**Figure 7 materials-16-01264-f007:**
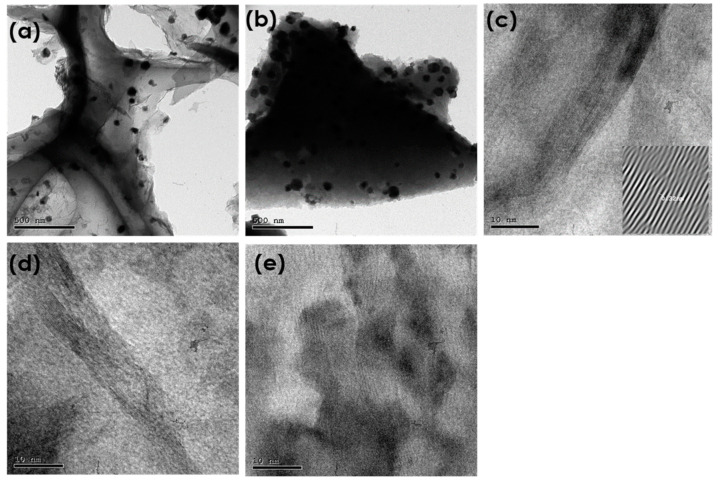
HRTEM images of (**a**) NiMo1/AC, (**b**) NiMo2/AC, (**c**) NiMo3/AC, (**d**) NiMo4/AC, and (**e**) NiMo5/AC.

**Figure 8 materials-16-01264-f008:**
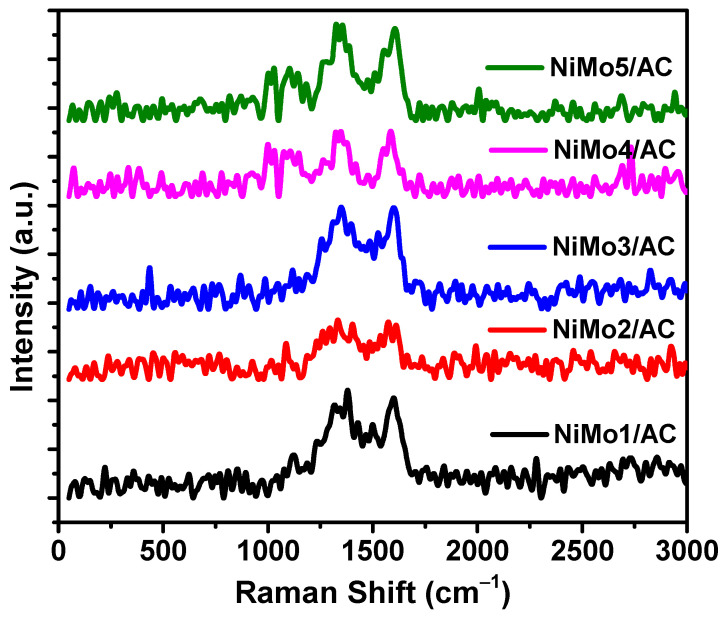
Raman analysis of the prepared samples.

**Figure 9 materials-16-01264-f009:**
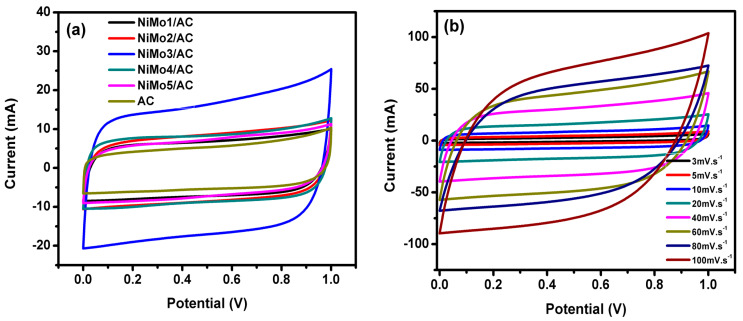
CV curves of (**a**) AC, NiMo1/AC, NiMo2/AC, NiMo3/AC, NiMo4/AC, and NiMo5/AC at a scan rate of 20 mV/s, and (**b**) NiMo3/AC at different scan rates.

**Figure 10 materials-16-01264-f010:**
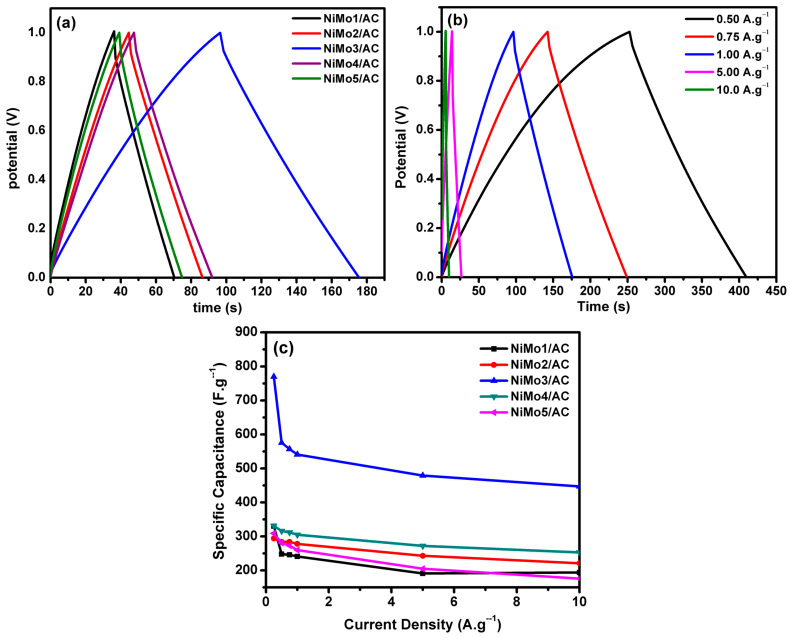
(**a**) The galvanostatic charge-discharge curves for all of the electrodes at a current density of 1 A·g^−1^, (**b**) the galvanostatic charge-discharge plots of NiMo3/AC at various current densities, (**c**) the change in specific capacitance with the applied current density.

**Figure 11 materials-16-01264-f011:**
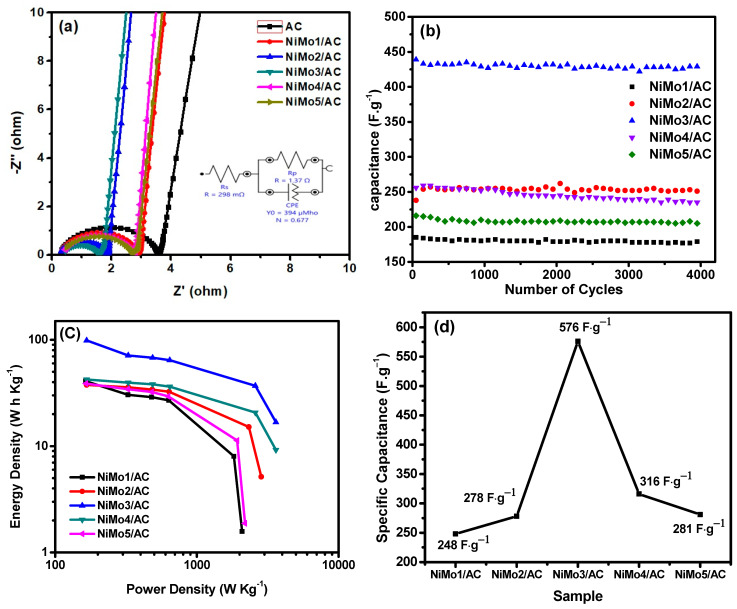
(**a**) EIS, (**b**) 4000th cyclic stability, (**c**) ragone plot, and (**d**) the change of specific capacitance with Ni-Mo ratio.

**Table 1 materials-16-01264-t001:** The abbreviations for the prepared samples.

Sample	Ration between		Abbreviation
	Ni	Mo	
Activated carbon	0	0	AC
NiMoO_4_/activated carbon	1	1	NiMo1/AC
	1	2	NiMo1/AC
	1	3	NiMo1/AC
	1	4	NiMo1/AC
	1	5	NiMo1/AC

**Table 2 materials-16-01264-t002:** Data obtained from the surface area analysis.

Sample	S_BET_	Pore Volume	Pore Diameter	Pore Volume (NLDFT)
	m^2^·g^−1^	cm^3^·g^−1^	nm	cm^3^·g^−1^
AC	1630	0.572	1.07	0.873
NiMo1/AC	1634	0.654	1.60	0.829
NiMo2/AC	1043	0.471	1.24	0.627
NiMo3/AC	1703	0.735	1.73	0.925
NiMo4/AC	1454	0.642	1.76	0.742
NiMo5/AC	1345	0.555	1.65	0.694

**Table 3 materials-16-01264-t003:** The specific capacitance values are derived from CV and GCD. Capacitance retention RCs after 4000 cycles at 4 A g^−1^. EIS is used to calculate the equivalent series resistance (ESR).

Sample	Ni	Mo	CV			GCD			ESR	Rs	RCs
			C^5mV/s^	C^10mV/s^	C^20mV/s^	C^0.5A/g^	C^0.75A/g^	C^1A/g^			
			F·g^−1^	F·g^−1^	F·g^−1^	F·g^−1^	F·g^−1^	F·g^−1^	Ω	Ω	%
AC	0	0	172	169	162	240	228	210	3.58	0.32	-
NiMo1/AC	1	1	197	196	189	248	246	241	2.91	0.30	96.75
NiMo2/AC	1	2	236	232	229	288	284	278	1.88	0.33	98.8
NiMo3/AC	1	3	454	438	418	576	557	541	1.60	0.41	97.7
NiMo4/AC	1	4	236	234	218	316	312	305	2.75	0.44	91.79
NiMo5/AC	1	5	203	196	189	281	273	260	2.81	0.50	94.91

**Table 4 materials-16-01264-t004:** Electrochemical behavior of NiMoO_4_ based on the carbon material in various studies.

Sample	Electrolyte	Substrate	Specific Capacitance F·g^−1^	RCs %	Reference
NiMo3/AC	1 M H_2_SO_4_	Graphite	541@1 A·g^−1^	97.7@4000cycle	This work
C@NiMoO_4_	2 M NaOH	Nickel foam	268.8@1 A·g^−1^	88.4@2000cycle	[28]
Mn_0.1_Ni_0.9_MoO_4_/rGO	KOH/PVA Gel Solid	Nickel foam	109.3@1 A·g^−1^	96.1@200cycle	[7]
NiMoO_4_/g-C_3_N_4_	6 M KOH	Carbon paper	510@1 A·g^−1^	91.8@2000cycle	[29]
NiMoO_4_/nanographite	1 M KOH	Graphite foils	454.5@0.5 A·g^−1^	72@1000cycle	[30]
NiMoO_4_/polyaniline	1 M Na_2_SO_4_	Nickel foil	285@5 mV·s^−1^	92@500cycle	[11]
WS_2_/α-NiMoO_4_	2 M KOH	Nickel foam	460@1 A·g^−1^	92@2000cycle	[31]

## Data Availability

The data are available upon request.

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
