# Peer review of "2D Hierarchical NiMoO4 Nanosheets/Activated Carbon Nanocomposites for High Performance Supercapacitors: The Effect of Nickel to Molybdenum Ratios"

_materials, 2023, doi:10.3390/ma16031264_

Round 1

Reviewer 1 Report

Although there is already research in this field, such as that carried out by Yuan, J., et al., and Thiagarajan, K., et al. The data obtained thanks to the relationships used between Nickel and Molybdenum of 1:3 yield novel results.

They show results in the present investigation of a high specific capacitance in comparison with other investigation works whose results are below the results shown in the present article.

In addition to the fact that the molar ratio between nickel and molybdenum of 1:3 generates great stability in the electrodes, we observe that the results obtained in this investigation are much better compared to previous investigations.

It is important to highlight that the method used (impregnation method) ended up giving optimal results in the homogeneous distribution of the elements.

Comparing the work shown in this article with the cited works, it is shown that the NiMo 1:3 ratio has a higher conductivity and a lower resistance.

The data shown in the article are supported by results obtained, resources used by other authors are cited, there is a clear relationship between what is stated and the data shown.

Due to the aforementioned, the results obtained, the methods used and the correctness of the information presented, I consider this article as an investigation with novel results.

1.- it is necessary to add more references.

2.- It is necessary to add a nomenclature table, place it as an annex.

4.- increase the size of the text of figure 5a,5b,5c,5d, 5e and 6.

5.- Figure 6b cannot be seen, it is necessary to find a way to further increase its size and highlight the axes more. Also differentiate figure 6b and the table in there, several images are presented but there are no subsections that mention what they refer to 6a, 6b and 6c .

6.- The discussion must be include future work, "Future research directions may also be mentioned."

7.- They conclude “These results suggest that there is an 325 optimum molar ratio between Ni and Mo in the synthesis of nickel molybdate nanoparti- 326 cles which greatly affect their performance in certain application” they should mention some of the application affect their performance

The article can be accepted after correcting the observations mentioned above.

Author Response

Dear Reviewer

Please find attached the answers to your comments.

Best Regards

Abdalla

Reviewer 2 Report

In this work, the author reported "2D Hierarchical NiMoO4 Nanosheets/Activated Carbon Nano- 2 composites for High-Performance Supercapacitors: The Effect of Nickel to Molybdenum Ratios" and systematically characterized using various physiochemical techniques. Therefore, I recommend this work for publication in the Materials journal. However, some of the major concerns should be addressed before proceeding with further actions.

1.     Authors should include Raman analysis for all the sample to evaluate the graphitic nature of the samples.

2.     The eqn.2 is wrong. Authors should check this.

3.     What is the mass loading of active materials?

4.     Why the surface area of the samples are differed with one another? Since the author used same activated carbon to produce composite.

5.     Authors should include BET analysis of the prepared activated carbon.

6.     There are many biosources available. Authors should include the reason for the selection of argan seed.

7.     Table three authors should correct .5 A/g and .75 A/g.

8.     Authors should include Rs and Rct values in the EIS analysis discussion.

9.     Authors should include an equivalent circuit fit diagram in Nyquist plot.

Author Response

(The authors gave the same response as above.)

Reviewer 3 Report

Current paper discussed how Nickel-Molybdenum ratio affects the efficiency of the electrode. Various samples were prepared and tested as supercapacitors electrodes. But some part of this work are not clear and some suggestion should be answer: 

1. Introduction is not enough discussed and should be improved.

2. The innovation of this paper is not highlighted. Explain clearly what the latest progress and previous work of this paper are based on?

3. Figure 1 was not good prepared, the resolution is not good. The authors mentioned two broad peak at 26 and 44 degree were obtaind, the first one is not clear if figure. Use wider one or detailed 26 degree in other figure.

4. what can be understand from narrow pick in figure 2.  Why NiMo3/AC have wider peak, the author should explain more in detailed. How many of this samples were tested. Does not the difference related to inaccurate experimental condition?

5. Graph of figure 6 is not clear. It should be rearrange. 

6. Dissimilar font in figures, formulas and text should be eliminate( formulas 1 and 2). Use proper capital letter to cite table 3.

Author Response

(The authors gave the same response as above.)

Round 2

Reviewer 2 Report

The authors have performed the revision work in a proper manner. Now this revised version of the manuscript can be published.

Reviewer 3 Report

The previous comments were completely done.